# Impact of COVID-19 Pandemic Constraints on the Ecobiochemical Status of Cultivated Soils along Transportation Routes

**DOI:** 10.3390/toxics11040329

**Published:** 2023-03-30

**Authors:** Elżbieta Zawierucha, Marcin Zawierucha, Barbara Futa, Agnieszka Mocek-Płóciniak

**Affiliations:** 1Department of Nursing, Midwifery and Emergency Medicine, Jan Kochanowski University in Kielce, IX Wieków Kielc 19A, 25-317 Kielce, Poland; 2Department of Agriculture and Rural Development, The Marshal Office of the Świętokrzyskie Voivodeship, IX Wieków Kielc 3, 25-516 Kielce, Poland; 3Institute of Soil Science and Environment Management, University of Life Sciences in Lublin, Leszczyńskiego St. 7, 20-069 Lublin, Poland; 4Department of Soil Science and Microbiology, Poznań University of Life Sciences, Szydłowska 50, 60-656 Poznan, Poland

**Keywords:** pandemic COVID-19, roadside cultivated soils, enzymatic activity, polycyclic aromatic hydrocarbons, heavy metals

## Abstract

There is a lack of studies on the impact of COVID-19-related population mobility and freight transport restrictions on the soil environment. The purpose of this study was to evaluate the impact of automotive pollution on selected parameters describing the quality and healthiness of crop soils based on results obtained before the pandemic (2017–2019) in relation to data from the pandemic period (2020–2021). The study included soils from six cultivated fields located in eastern Poland along national roads (DK No. 74 and 82) and provincial roads (DW No. 761 and 835). Soil samples were taken from distances of 5, 20, 50, and 100 m from the edge of the roadway. The following soil characteristics were determined: pH_KCl_, content of total organic carbon (TOC), total nitrogen (TN), and activity of the three enzymes dehydrogenases (ADh), neutral phosphatase (APh), and urease (AU). The degree of traffic-generated soil pollution was assessed by determining the samples’ total cadmium and lead levels (Cd and Pb) and total content of 14 polycyclic aromatic hydrocarbons (Σ14PAHs). The monitoring of cultivated soils showed that the parameters of cultivated soils varied primarily according to the distance from the edge of the roadway. There was an increase in soil acidity and TOC and TN content and a decrease in Cd, Pb, and Σ14PAHs as one moved away from the edge of the roadway. The highest ADh and APh values were found in soils located 100 m from the edge of the road. AU at 5 m and 20 m from the edge of the pavement was significantly higher than at 100 m away. The reduction in vehicular traffic associated with the pandemic did not affect the changes in the reaction of the studied soils and their TOC, TN, and Pb contents. The lowest content of Σ14PAHs was found in 2020. In the case of the amount of Cd in soils, a downward effect was also observed in 2020. However, no significant differences were noted, except for the soils in Skorzeszyce and Łuszczów Kolonia. The reduced influx of xenobiotics into the soil environment stimulated ADh and APh. In the following year (2021), the amounts of tested xenobiotics and enzyme activities in the soils were at a similar level to those in 2019. The results indicate a positive but short-term effect of the pandemic on reducing the contamination of soils located along transportation routes.

## 1. Introduction

The COVID-19 pandemic has caused a major upheaval that has triggered socio-political changes around the world. The first media reports of illness caused by a previously unknown type of coronavirus, SARS-CoV-2, appeared early in December 2019 in the city of Wuhan in central China. In just 30 days, it spread from one city to all of China and then appeared in other countries. In the second half of February 2020, outbreaks of SARS-CoV-2 infections occurred in Europe (mainly in Italy), and as of 4 March 2020, infections were reported in Poland, eastern Europe [1,2,3]. On 30 January 2020, the World Health Organization (WHO) declared the COVID-19 outbreak caused by the SARS-CoV-2 coronavirus a public health emergency of international concern. On 11 March 2020, the WHO declared the COVID-19 outbreak a pandemic [4]. In the spring of 2020, in order to counter the spread of infection, restrictions were imposed in most countries, including restrictions on movement; temporary total or partial quarantine; and the postponement or cancellation of a number of sports, religious, and cultural events. At the national and local levels, schools and universities were closed. Some countries closed borders or imposed restrictions on border traffic. The introduced restrictions resulted in a reduction in emissions from certain sectors of economic activity (including transportation and selected industries) and a potential increase in emissions from municipal and residential sources. This was due to the population remaining in quarantine in households, as well as widespread work and education from home using digital technologies [1,3,5]. In the lockdown period, traffic decreased by 50 to 70% and as much as 90% in some cases [6].

A number of publications have been published on the effects of COVID-19 restrictions on environmental pollutant concentrations, and the vast majority have focused on air quality [3,7,8]. However, there is a lack of studies describing parameter changes in soils [9]. Soils are the foundation on which all life on Earth rests, and their ecosystem functions are an indispensable part of the food-production process [10]. The ecochemical state of soils determines crop quality and human health [11,12]. However, between 60 and 70% of EU soils are unhealthy according to a report on a Horizon Europe mission on soil health and food [13]. According to the Intergovernmental Technical Panel on Soils (ITPS), the third most significant threat to soil function in Europe is pollution from anthropogenic activities [14]. Particularly noteworthy are cultivated soils located along transportation routes. Road transport and automobile communications are an important source of heavy metal and polycyclic aromatic hydrocarbon (PAH) pollution of soils adjacent to transportation routes. Vehicle emissions, abrasion of vehicle tires and asphalt, wear and tear of the clutch and brake pads and discs, and leakage of operating fluids are important sources of these pollutants in the soils around roads [15,16,17]. Soils contaminated with heavy metals (HMs) and PAHs can pose a health risk to humans, livestock, and wildlife, as well as an ecotoxicological risk to the soil biome, soil functions, plants, the quality of air, and aquatic life [18,19].

Intensive development of motorization has generated certain threats to the quality of the soil environment, human health, and life. In this context, scientifically designed and continuous monitoring of agricultural land along roads is necessary. The protection and monitoring of these soils are in line with the environmental policy of EU member states, which aims to achieve the goals of the European Green Deal [20]. The determination of the content of xenobiotics (i.e., HMs or PAHs) in soils does not reflect the real ecotoxicological risk associated with their presence in the environment. Therefore, when assessing the quality of soils, what is important is the amount of contamination that can be tolerated and without causing negative effects on living organisms [21,22].

A reliable assessment of the ecochemical state of soil can be achieved by testing the activity of a number of enzymes, including dehydrogenases, urease, and neutral phosphatase. Soil enzymes are natural mediators and catalysts of many important soil processes [23] and participate in the formation and decomposition of organic matter in the soil. Enzymes in the soil also stimulate reactions that release nutrients from organic matter and make them available to plants. They have a significant effect on the rate of transformation and flow of carbon, nitrogen, and many other elements of the biochemical cycle and also stimulate the fixation of molecular nitrogen [24]. Enzymatic activity is a sensitive indicator of changes in the soil environment, including those caused by the presence of HMs and PAHs in the environment [22,25,26,27].

The purpose of this study was to assess the impact of population mobility and freight transport restrictions associated with the COVID-19 pandemic and the distance from the edge of a roadway on the ecobiochemical status of soils in cultivated fields. The fields were located along transportation routes in the Świętokrzyskie and Lubelskie provinces of Poland. An analysis of the results obtained before the pandemic (2017–2019) in relation to data from during the pandemic period (2020–2021) allowed us to assess the impact of automotive pollution on selected parameters describing the quality and health of crop soils. The present study is a continuation of research previously conducted in the mentioned areas in 2017–2019 [28].

## 2. Materials and Methods

### 2.1. Study Area

The study included soils from six farm fields located in the eastern part of Poland along national roads (DK Nos. 74 and 82) and provincial roads (DW Nos. 761 and 835) in the Świętokrzyskie and Lubelskie provinces (Figure 1). The public roads in Poland are classified according to their function in the road network as national roads, which are transit routes, and provincial roads, which are connections between cities of importance in the province. With the spread of the SARS-CoV-2 coronavirus in Europe, restrictions on the mobility of the population and the transport of goods have been imposed since March 2020. The study was conducted in 2020–2021, and the results were compared with data from 2017–2019, which were the basis of a doctoral thesis [28] and article [29].

The soils in the studied area were classified as Cambisols and Luvisols and characterized by silt loam (SiL) and silt (Si) texture (Table 1) [30,31]. Field work was conducted during periods of stable weather, when the soil was in a state of dynamic equilibrium, which kept the course of biochemical processes within the limits of moderate intensity.

### 2.2. Sampling and Analyses

Soil samples were taken from selected agricultural fields in August of each research year (2017–2021) from the arable layer at depths of 0–20 cm and distances of 5, 20, 50, and 100 m from the edge of the roadway. Soil samples were taken in triplicate using a special cane with a diameter of 4 cm. Soil samples were collected for the biochemical analyses, sieved through a 2 mm sieve, and stored at 4 °C according to the principles specified in ISO 18400 [32]. Soil samples for physicochemical analyses were dried at room temperature and then ground in a soil mill. Each sample was assayed using three replications.

The physicochemical analyses consisted of the determination of the following parameters: pH_KCl_, content of total organic carbon (TOC), and total nitrogen (TN). To assess the degree of traffic-generated soil pollution, total cadmium and lead (Cd and Pb) and the total content of 14 polycyclic aromatic hydrocarbons (Σ14PAHs) were determined in the collected samples. pH_KCl_ was determined by the potentiometric method in a 1-mol·dm^−3^ KCl (1:2.5) solution [33]. TOC was determined by the Tiurin method [34] through the combustion of soil samples using a TOC-VCSH apparatus with an SSM-5000A module (Shimadzu Corp., Kyoto, Japan). TN was determined by the modified Kjeldahl method using a Kjeltech TM 8100 distillation unit (Foss, Copenhagen, Denmark) [35]. The total content of HMs (Cd and Pb) was determined using inductively coupled plasma atomic emission spectroscopy (ICP-AES) on an optical emission spectrometer (PS 950 ICP-OES, Teledyne Leeman Labs, Hudson, NY, USA) after ashing the soil at 450 °C and digesting it in an aqua regia solution (HCl-HNO_3_ at a 3:1 ratio). Determination of Σ14PAHs was performed by an HLPC method on a liquid chromatograph (ThermoSeparation Product, Waltham, MA, USA) with UV detection (254 nm) [36].

Biochemical analyses were conducted to determine the activity of the following soil enzymes: dehydrogenases (EC 1.1), neutral phosphatase (EC 3.1.3), and urease (EC 3.5.1.5). These enzymes catalyze the carbon (dehydrogenases), nitrogen (urease), and phosphorus (neutral phosphatase) cycling processes in ecosystems. The methodology for determination of the soil enzymatic activity was based on a detailed study conducted by Schinner et al. [37] and Dick [38]. Table 2 shows the classification of the soil enzymes tested (EC); their acronyms, substrates, and products used in the assays; and the units used for analytical data.

The activity of dehydrogenases (ADh) was determined by Thalmann’s method [37] using a 1% solution of 2,3,5-triphenyl tetrazolium chloride (TTC) as a substrate. The determination of neural phosphatase activity (APh) was performed according to Tabatabai and Bremner [37] using a 0.8% disodium p-nitrophenyl phosphate solution as a substrate in pH 6.5 buffer. Urease activity (AU) was determined using Zantua and Bremner’s method [37] and a 2.5% urea solution as a substrate. The activities of the enzymes were determined by the colorimetric method with a CECIL CE 2011 spectrophotometer (Cecil Instrumentation Ltd., Cambridge, UK) at the following wavelengths: λ = 485 nm for dehydrogenases, λ = 410 nm for urease, and λ = 410 nm for neutral phosphatase.

### 2.3. Statistics

A statistical analysis of the results was carried out using Microsoft Office Excel 2010 and the package Statistica 14.1 PL (TIBCO Software Inc., Palo Alto, CA, USA). Descriptive statistics included arithmetic means and standard deviation for individual parameters. A statistical evaluation of the variability of the results was performed using a two-factor analysis of variance. The significance of differences between mean values was verified using Tukey’s HSD post hoc test at a significance level of *p* ≤ 0.05. The Shapiro–Wilk test was used to assess the normality of the data. The value of Pearson’s linear correlation coefficient (r) was calculated for selected parameters with a significance level of *p* < 0.05. A maximum of 5% scatter between measurements in the chemical analysis was assumed in the study. In addition, regression models were determined between the tested soil properties and (I) the distance from the road edge and (II) changes in the mobility of the population and the transport of goods in 2017–2021.

## 3. Results

### 3.1. Reaction of the Studied Soils

The pH_KCl_ values of soils varied from 3.86 to 7.28, indicating a wide range of pH from very acidic to alkaline (Table 3). An increase in the acidity of soils was observed as one moved away from the edge of the roadway. In soils close to the road (5 and 20 m), pH_KCl_ values were generally significantly higher than at a distance of 100 m (within the range of 0.52–2.64 units in 1 mol KCl dm^−3^). During the study period, there were no statistically significant differences in pH_KCl_ values (Table 3).

### 3.2. Organic Carbon and Total Nitrogen Content

The analyzed sites varied in TOC and TN content. They contained from 6.38 to 10.28 gTOC kg^−1^ and 0.42 to 1.15 gTN kg^−1^. According to the criteria of the European Soil Bureau Network (ESBN), the tested soils were generally characterized by very low TOC content [39]. Soils located at a distance of 5 m from the edge of the road had the lowest TOC and TN contents. The values of these parameters increased (statistically significantly in general) with the distance from the bituminous pavement (Table 4 and Table 5). At a distance of 100 m from the edge of the roadway, TOC levels were about 14–42% higher and TN levels were about 25–86% higher than in the immediate vicinity of the bituminous pavement. As in the case of pH_KCl_, there was no statistically significant variation in the content of TOC and TN (Table 4 and Table 5) during the analyzed period.

### 3.3. Cadmium and Lead Content

Cd and Pb contents in the studied soils ranged from 0.25 to 0.63 mgCd kg^−1^ and from 13.8 to 34.9 mgPb kg^−1^ (Table 6 and Table 7). The factors significantly modifying cadmium and lead contents in the studied soils were the distance from the edge of the roadway. The highest Cd and Pb contents were recorded in soils at distances of 5 m and 20 m from the edge of the roadway. They decreased statistically significantly at 50 m and 100 m from the road.

There was a decrease in the amount of Cd content in 2020 (the year of major COVID-19 restrictions) and then an increase in 2021. However, the recorded differences were generally not statistically significant except for the soils in Skorzeszyce and Łuszczów Kolonia (Table 6). In the case of Pb, the highest content in soils was recorded in 2017, followed by a subsequent decrease, but the differences were not statistically significant (Table 7).

### 3.4. Content of PAHs

The contents of Σ14PAHs in the studied soils ranged from 349.4 to 1328.6 μgkg^−1^ (Table 8). The modifying factors of Σ14PAH contents in the studied soils were the distance from the edge of the roadway and years of study. The highest Σ14PAHs contents were recorded in soils at 5 m from the edge of the roadway (Table 8). In soils located 50 m and 100 m from the edge of road, Σ14PAHs levels were more than 50% lower than in soils in the immediate vicinity of asphalt roads (5 m and 20 m).

After analyzing changes in Σ14PAHs in the analyzed soils from 2017 to 2021, the lowest content of these xenobiotics was found in the year of the lockdown due to the COVID-19 pandemic, i.e., 2020. This effect was most pronounced in the soils of the observation plots located at distances of 5 and 20 m from the edge of the roadway. For most sites, except Skorzeszyce and Albertów, these differences were statistically significant (Table 8). In 2021, the content of Σ14PAHs in the investigated soils increased and was at a similar level to 2019. The results indicate a short-term effect of the pandemic on the reduction of contamination by PAHs of soils located along traffic routes.

### 3.5. Enzymatic Activity

ADh, APh, and AU showed pandemic-related variations according to the type of enzyme, location, distance from the edge of the roadway, and years of study (Table 9, Table 10 and Table 11).

ADh ranged from 0.80 to 4.85 mg TPF kg^−1^ 24 h^−1^ (Table 9). The distance from the edge of the roadway significantly affected the activity of these enzymes. The highest ADh values were determined in soils located 100 m from the road. In 2020, an increase in ADh was found compared to other years, with statistically significant differences noted for only the Piekoszów and Skorzeszyce sites (Table 9).

APh ranged from 5.64 to 14.56 mmol PNP kg^−1^ h^−1^ (Table 10). The distance from the edge of the roadway significantly affected the activity of these enzymes. The highest APh values were found in soils 100 m from the edge of the road. In the year of the outbreak of the pandemic in Europe (2020), there was a significant increase in APh compared to other years. Only the Marcinkowice site did not show statistically significant differences (Table 10).

AU ranged from 8.62 to 17.40 mg N-NH_4_^+^ kg^−1^ h^−1^ (Table 11). The distance from the edge of the roadway significantly affected the result. AU at 5 m and 20 m from the edge of the pavement was significantly higher than at 100 m away. Restrictions on vehicular transport associated with epidemic restrictions had a different effect on the direction and severity of changes in AU than ADh and APh. In 2020–2021 (under the conditions of reduced input of automotive pollutants into the soil environment), AU was lower than in 2017–2019 (Table 11). However, significant differences were observed only in the Marcinkowice and Skorzeszyce locations.

## 4. Discussion

Technological development and globalization are significantly influencing the expansion of transportation routes, and the rise in people’s standard of living has intensified transportation by passenger cars [40]. The negative impact of automobile transportation on soils, including agricultural soils, is associated with environmental pollution by HMs and PAHs, among others [22,41,42]. The protection and monitoring of agricultural soils, including agricultural land along roads, are related to the European Union’s broad environmental policy. Healthy agricultural soils and cleaner, more sustainable transportation are key elements to achieving the European Green Deal goals of climate neutrality, restoration of biodiversity, healthy and sustainable food systems, and a resilient environment [20].

### 4.1. Effect of Distance from Road Edge

The values of physicochemicals (pH_KCl_, TOC, and TN), biochemicals (ADh, AU, and APh), and Σ14PAHs, Cd, and Pb varied over a wide range and primarily according to the distance from the edge of the road (Table 2, Table 3, Table 4, Table 5, Table 6, Table 7, Table 8, Table 9, Table 10 and Table 11). This was confirmed by the multivariate regression analysis (Figure 2).

During the study period, an increase in soil acidity was observed as one moved away from the edge of the roadway. In soils close to it, pH_KCl_ values were significantly higher than at 100 m from the road. A similar relationship was shown by other studies [43,44]. According to them, the alkalinization of soils near roads is related to alkaline dust precipitation and the use of slip-control agents in winter.

Werkenthin et al. [45] showed that the range of impact of alkalizing deposition on the soil environment from pavement abrasion and brake linings does not exceed 5–10 m. However, Cunningham et al. [43] noted elevated sodium contents as far as 150 m from paved surfaces. According to Green et al. [46] and Gavrichkova et al. [44], 20–90% of applied road salts enter the roadside environment through direct infiltration, surface runoff, spray or aerosol fallout, and flushing from vegetation. This causes changes in the saturation of the sorption complex with base cations, disrupting natural biogeochemical cycles, and consequently raising the pH of soils [44,46]. The accumulation of base cations depends primarily on the content of clay particles in the soil and their associated negative charges [43,47], and it has been observed that reduced infiltration is a common phenomenon in roadside soils [48].

TOC and TN contents in soils increased with distance from the edge of the road, and the change was statistically significant in general. The influx of automotive pollutants and increased salinity of the soil environment had a significant effect on the decrease in TOC and TN contents in soils near roads compared to sites 100 m from roads. The contaminants of roadside soils (mainly chloride salts, HMs, and PAHs) negatively affected the processes of carbon and nitrogen accumulation and their transformation in the soil environment. They reduced the abundance, diversity, and activity of soil microorganisms and stimulated the solubility of soil organic matter and thus its mineralization and potential gas or leaching losses [49,50]. The negative correlation demonstrated between pHKCl and TOC content (r = −0.61) and TN content (r = −0.49) indirectly confirms the presence of this mechanism.

The soil TOC reservoir had a decisive effect on the TN pool (r = 0.97) (Table 12). The decrease in soil organic matter undoubtedly also had an effect on the decrease in TN content in roadside soils [50]. Changes in pH due to soil salinity near roads in the long term may also affect the rate of key microbial N-transformation processes (ammonification and nitrification) by slowing or even inhibiting them [46,51]. The long-term effects of winter slip-control agents also exacerbate leaching of nitrate (V) and dissolved organic nitrogen (DON) compounds from soils in the vicinity of roads [46,48].

The highest Cd and Pb contents in soils were found in the immediate vicinity of the roadway (5 m and 20 m). The amounts of HM in soils decreased significantly with increasing distance from the road. Other studies on soils subjected to vehicular transport pressure also observed a decrease in the amount of HM with the distance from transportation routes [22,29,45,52,53]. According to Krailertrattanachai et al. [53], a safe distance for agricultural production should be greater than 10 m from the edge of the road. In contrast, da Silva et al. [50] analyzed literature data and found that most metals are deposited up to 20 m from the edge of the roadway, with the possibility of dispersion up to 100–200 m away. This depends on the terrain, traffic volume, rainfall totals, and wind speed and direction [50,52,54].

Trace elements of the automotive origin are more mobile in the roadside zone, which is primarily due to their interaction with salts used for slush-slippage control and the activation of mechanisms related to the ion exchange, formation of complexes with chlorides, and dispersion of colloids [55]. Elevated salinity can also lead to the release of previously adsorbed HMs and enhance their transport to the surrounding area [56,57]. HMs accumulating in the roadside belt or moving into agroecosystems are extremely dangerous due to the possibility of their secondary activation and bioaccumulation in human, plant, and animal tissues [58,59]. At the same time, it should be emphasized that the determined contents of Cd and Pb did not exceed the lowest permissible contents, causing risks of particular importance for the protection of the Earth’s surface specified in the Regulation of the Minister of Environment of 1 September 2016 [60]. They can be used for all horticultural and agricultural crops in accordance with the principles of rational use of agricultural production space.

The highest Σ14PAH contents in soils were recorded at 5 m from the edge of the roadway, decreasing significantly with distance. Other studies also reported significantly higher PAH contents in soils directly influenced by transportation [22,61,62]. Futa et al. [22] and Yang et al. [63] noted that ΣPAH contents in agricultural soils decreased exponentially with increasing distance from transportation routes. On the other hand, Zhang et al. [62] showed that Σ16PAH contents in agricultural soils at 0 m and 20 m points from the edge of the roadway were similar to each other. From 20 m to 50 m, Σ16PAH amounts increased with distance, while from 50 m to 250 m, Σ16PAH concentrations gradually decreased with increasing distance, reaching a minimum value at 250 m. PAH concentrations at 0 m and 20 m were lower than those at 50 m, which was because the foliage of the roadside trees in spring hindered the diffusion of pollutants [62].

The highest concentration at 50 m is mainly due to the transfer of particulate matter and aerosols generated by vehicle exhaust and the impact of runoff after heavy rainfall [64,65]. The PAH concentrations in soils are not only related to their input from exogenous sources, but also to soil properties, including organic carbon content, nitrogen content, and pH. These factors affect the losses, including sorption and desorption, transport, leaching, volatilization, biological uptake, and decomposition [66]. However, our study showed no correlation between Σ14PAHs and pHKCl, TOC, and TN (Table 12). The highest amounts of Σ14PAHs, cadmium, and lead accumulated in the immediate vicinity of the edge of roadways, indicating the need to monitor and protect this area of agroecosystems from the potential negative impacts of traffic routes.

The distance from the edge of the roadway significantly affected soil enzyme activity, and the direction and intensity of changes depended on the type of enzyme. The highest values of ADh and APh were found in soils located 100 m from the edge of the road. The reason for the impaired activity of ADh and APh in soils located in the immediate vicinity of traffic routes was the increased influx of automotive pollutants and salt into the soil environment for de-icing roads. Salinity as well as HMs and PAHs contained in exhaust gases and road dust negatively affect the abundance and activity of microorganisms, disrupting their basic physiological functions and primary processes related to the decomposition and transformation of organic matter [67,68].

It is well known that soil microorganisms are mainly responsible for the biosynthesis of soil enzymes [69,70]. Data presented by many studies confirm the particular sensitivity of dehydrogenases and phosphatases to HM and PAH contamination of the soil environment [22,26]. Their activity can be used as an indicator of soil environment contamination with these xenobiotics [71,72,73].

In contrast, AU at 5 m and 20 m from the edge of the pavement was significantly higher than at 100 m away. The pH_KCl_ of the soils developed in a similar way. According to previous research, urease activity is usually positively correlated with soil pH [74,75]. In this study, the correlation analysis performed did not show such a relationship.

### 4.2. Impact of Traffic Restrictions Caused by COVID-19 Pandemic

Traffic and automobile transport restrictions caused by the COVID-19 pandemic did not significantly affect the pH, TOC, and TN (Table 3, Table 4 and Table 5). Under the conditions of cultivated fields, the factors modifying pH, TOC, and TN may have been agrotechnical treatments. The multivariate regression analysis did not show significant correlations between the soil parameters and the years of the study [76]. In addition, the multiple regression analysis carried out did not show any significant changes in the tested soil parameters during the research period (2017–2021).

Between 2017 and 2021, the lowest contents of Σ14PAHs were found in 2020, the year with the greatest COVID-19 restrictions. In the discussed year, a decrease (not statistically significant) in the amount of Cd in soils was also observed. This effect was most pronounced at 5 and 20 m from the edge of the roadway. In 2021, the contents of these xenobiotics increased and were at similar levels to those in 2019. The results indicate a short-term effect of the pandemic on the reduction of contamination by PAHs, and, to a lesser extent, the Cd of soils located along traffic routes.

The results are supported by data published by the Polish Organization of Petroleum Industry and Trade (POPiHN), which is associated with 56% of fuel stations in Poland [77,78]. These data show that in 2020, there was a slump in liquid fuel sales related to the COVID-19 pandemic, with a 35% drop in April 2020 compared to April 2019. For the whole year, the average decrease in the growth rate of fuel sales at stations was 8.4% (8% for diesel, 8% for gasoline, and 12% for autogas) [77]. In 2021, there was a recovery of the fuel market in Poland and worldwide. Sales of gasoline increased by 11% compared to the previous year, while diesel sales increased by 7%, and autogas sales increased by 2% [78]. However, there was a downward trend in the amount of Pb, which may have been related to a reduction in emissions of Pb compounds from exhaust gases. Industrial incineration processes, coal combustion, and petrol-powered vehicles are the main sources of the PAHs and Cd, while Pb mainly originates from historical accumulation and the use of Pb-enriched petrol [79]. The introduction of tetraethyl lead and tetramethyl lead as an agent to increase the acetate number and prevent engine knocking has made the commute routes the main source of these metals [80,81].

Despite the fact that lead fuels have been unavailable in Poland since 2005, the accumulation of the metal in earlier years was so high that even limiting its content in fuel did not result in a significant decrease in its concentration in the soil. This is due to the fact that lead is not very mobile in soils, and at pH > 6.5, it becomes immobilized [80]. The correlation analysis showed a positive correlation between Σ14PAHs and Cd (r = 0.75), and a negative one for Σ14PAHs with Pb (r= −0.47).

In the year of the outbreak in Europe (2020), there was an increase in ADh and Aph compared to other years. The greatest differences were noted near transportation routes. The reason was the reduced influx into the soil environment of PAHs contained in exhaust fumes and road dust, which inhibit the biosynthesis of enzymes by soil microorganisms. Our study showed a negative correlation between ADh and Σ14PAHs (r = −0.40). The significance of dehydrogenases as a pollution indicator is supported by their lack of ability to accumulate in the extracellular environment since they are found in only living intact cells [82]. In 2021, the activity of dehydrogenases and neutral phosphatase was at a similar level to that in 2017–2019. This was influenced by the increase in vehicles on the roads and the associated increased influx of PAHs into the soil environment. The obtained results show that the reduction in road pollution triggers an immediate, positive reaction of the soil environment.

Restrictions on automobile transport associated with epidemic restrictions had a different effect on the direction and severity of changes in AU compared to ADh and APh. In 2020–2021 (with reduced automotive pollutants), AU was lower than in 2017–2019. Our study showed a positive correlation between AU, Cd (r = 0.72), and Σ14PAHs (r = 0.40) (Table 12). Urease is resistant to external factors and an increase in its activity is observed even in extreme conditions [83]. The high resistance of urease to anthropogenic pollutants (HMs and PAHs) was also shown by other studies [22,26]. Based on AU, the degree of anthropogenization of the soil environment can be assessed [84].

The activity of soil enzymes is also influenced by the species composition of the plant cover [23]. According to many authors [85,86,87], root secretions are a good source of nutrients for microorganisms, mainly living in the rhizosphere. During growth, roots produce organic and inorganic compounds, as well as active substances, which foster the growth of many enzyme-producing microbes. By creating an artificial simplified rhizosphere, Renella et al. [86] demonstrated that root secretions, such as glucose, glutamic acid, citric acid, and oxalic acid have a significant effect on the activation of microbial development. Root secretions affect both the growth of soil microorganisms and their adaptation to the decomposition of contaminants, which is particularly important in areas transformed by man [87]. However, it should be remembered that the effect of higher plants on soil enzymes depends on the chemical composition of the plant, which, even in the case of root secretions alone, may be different in different types, species, and even varieties [23,88]. The research shows that the assessment of soil quality and health should include tests of enzymatic activity. Determining the content of xenobiotics in soil does not reflect the real ecotoxicological risk associated with their presence in the environment. However, soil enzymes reflect the level of environmental contamination that threatens living organisms without identifying multiple compounds [22,87,89].

## 5. Conclusions

This study monitored cultivated soils along transportation routes in eastern Poland in 2017–2021, and the results showed that the values of pH_KCl_, TOC, TN, ADh, AU, APh, Σ14PAHs, Cd, and Pb primarily varied with distance from the edge of the roadway. Soil acidification increased as one moved away from the edge of the roadway. The highest amounts of Cd, Pb, and Σ14PAHs accumulated at a distance of 5–20 m from the edge of the roadway, indicating the need to monitor and protect this area of agroecosystems from the potential negative impacts of traffic routes.

The analysis showed that the reduction in motorized transport did not affect changes in the studied soils or their organic carbon, total nitrogen, and lead content. The lowest content of Σ14PAHs, and, to a lesser extent, Cd was found in 2020. The reduced influx of xenobiotics into the soil environment inhibited the enzyme biosynthesis by soil microorganisms and stimulated ADh and APh. In the following year (2021), the amounts of xenobiotics and enzyme activities were similar to levels in 2019. The obtained results show that the reduction in road pollution triggers an immediate, positive reaction of the soil environment. The impact of the restrictions related to the COVID-19 pandemic on the chemical and enzymatic properties of soils is noteworthy, even if it was too short to bring measurable environmental effects. The conducted research points to the need to reduce pollution from the transport sector. This is extremely important because both healthy soils and sustainable transport are among the 17 Sustainable Development Goals in the 2030 Agenda.

## Figures and Tables

**Figure 1 toxics-11-00329-f001:**
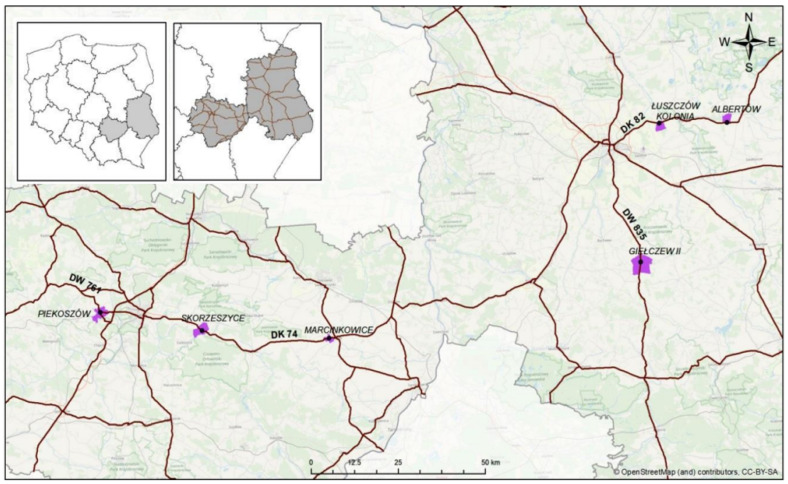
Location of the study area [28,29].

**Figure 2 toxics-11-00329-f002:**
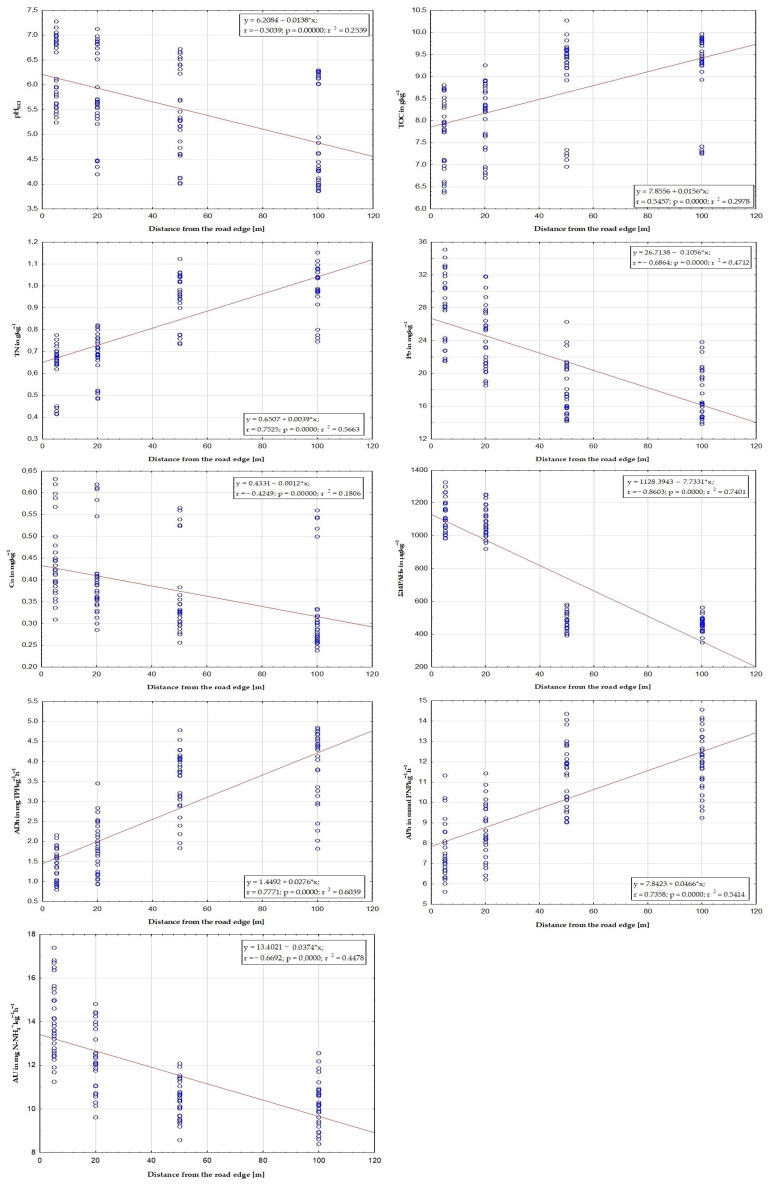
Regression analysis between studied soil parameters and distance from the road edge.

**Table 1 toxics-11-00329-t001:** Area inventory, location, soil types, and species.

Site	Road No.	Soil Type—Particle Size Group
Piekoszów	DW 761	Cambisols—silt loam (SiL)
Marcinkowice	DK 74	Cambisols—silt loam (SiL)
Skorzeszyce	DK 74	Luvisols—silt (Si)
Giełczew II	DW 835	Cambisols—silt loam (SiL)
Łuszczów Kolonia	DK 82	Cambisols—silt loam (SiL)
Albertów	DK 82	Luvisols—silt (Si)

Explanations: DK—national road; DW—provincial road.

**Table 2 toxics-11-00329-t002:** Determination of the activity of soil enzymes.

Enzymes	EC	Acronym	Substrate Name	Product Name	Unit Name
Dehydrogenases	EC 1.1	ADh	2,3,5-triphenyltetrazolium chloride (TTC)	triphenyl formazane (TPF)	mg TPF kg^−1^ 24 h^−1^
Neutral phosphatase	EC 3.1.3	APh	*p*-nitrophenyl phosphate disodium	*p*-nitrophenol (PNP)	mmol PNP kg^−1^ h^−1^
Urease	EC 3.5.1.5	AU	Urea	N-NH_4_^+^	mg N-NH_4_ kg^−1^ h^−1^

**Table 3 toxics-11-00329-t003:** The pH_KCl_ values in soils before the COVID-19 pandemic (2017–2019) [28] and during the pandemic (2020–2021).

Site	Years	Distance from the Road Edge [m]			
		5	20	50	100
Piekoszów LDS_0.05 for distance_ = 0.239 LDS_0.05 for years_ = n.s.	2017	6.81 ± 0.06 ^a^	6.74 ± 0.5 ^ab^	6.52 ± 0.01 ^bc^	6.29 ± 0.03 ^cd^
	2018	7.06 ± 0.02 ^a^	6.82 ± 0.4 ^ab^	6.64 ± 0.02 ^bc^	6.27 ± 0.03 ^d^
	2019	6.98 ± 0.02 ^a^	6.92 ± 0.3 ^a^	6.55 ± 0.01 ^b^	6.23 ± 0.03 ^bc^
	2020	6.66 ± 0.03 ^a^	6.51 ± 0.1 ^a^	6.42 ± 0.02 ^a^	6.02 ± 0.02 ^b^
	2021	6.99 ± 0.04 ^a^	6.88 ± 0.2 ^a^	6.51 ± 0.02 ^b^	6.27 ± 0.03 ^bc^
Marcinkowice LDS_0.05 for distance_ = 0.279 LDS_0.05 for years_ = n.s.	2017	6.82 ± 0.03 ^a^	6.82 ± 0.2 ^a^	6.72 ± 0.02 ^b^	6.12 ± 0.01 ^b^
	2018	6.84 ± 0.04 ^a^	6.98 ± 0.1 ^ab^	6.67 ± 0.01 ^b^	6.26 ± 0.02 ^d^
	2019	6.91 ± 0.02 ^a^	7.13 ± 0.2 ^a^	6.23 ± 0.02 ^b^	6.14 ± 0.03 ^bc^
	2020	6.75 ± 0.02 ^a^	6.63 ± 0.2 ^ab^	6.40 ± 0.02 ^b^	6.01 ± 0.02 ^c^
	2021	6.92 ± 0.03 ^a^	6.86 ± 0.2 ^a^	6.32 ± 0.01 ^b^	6.19 ± 0.03 ^bc^
Skorzeszyce LDS_0.05 for distance_ = 0.372 LDS_0.05 for years_ = n.s.	2017	5.55 ± 0.05 ^a^	5.32 ± 0.3 ^a^	5.71 ± 0.03 ^b^	4.28 ± 0.02 ^c^
	2018	5.61 ± 0.03 ^a^	5.66 ± 0.2 ^a^	5.69 ± 0.03 ^a^	4.32 ± 0.03 ^b^
	2019	5.86 ± 0.05 ^a^	5.67 ± 0.3 ^ab^	5.31 ± 0.02 ^b^	4.46 ± 0.02 ^c^
	2020	5.47 ± 0.06 ^a^	5.21 ± 0.2 ^a^	5.09 ± 0.01 ^a^	4.12 ± 0.02 ^b^
	2021	5.77 ± 0.04 ^a^	5.54 ± 0.1 ^a^	5.16 ± 0.01 ^a^	4.37 ± 0.02 ^c^
Giełczew II LDS_0.05 for distance_ = 0.243 LDS_0.05 for years_ = n.s.	2017	5.35 ± 0.01 ^a^	4.49 ± 0.4 ^bc^	4.04 ± 0.01 ^c^	4.28 ± 0.02 ^c^
	2018	5.41 ± 0.02 ^a^	4.35 ± 0.3 ^b^	4.01 ± 0.02 ^c^	4.09 ± 0.02 ^c^
	2019	5.63 ± 0.02 ^a^	4.47 ± 0.1 ^b^	4.12 ± 0.02 ^cd^	4.26 ± 0.04 ^d^
	2020	5.25 ± 0.02 ^a^	4.20 ± 0.1 ^b^	4.14 ± 0.02 ^b^	3.99 ± 0.02 ^bc^
	2021	5.53 ± 0.02 ^a^	4.48 ± 0.2 ^b^	4.02 ± 0.01 ^c^	3.87 ± 0.03 ^c^
Łuszczów Kolonia LDS_0.05 for distance_ = 0.258 LDS_0.05 for years_ = n.s.	2017	7.01 ± 0.03 ^a^	5.71 ± 0.2 ^b^	5.29 ± 0.02 ^c^	4.62 ± 0.02 ^d^
	2018	7.06 ± 0.03 ^a^	5.65 ± 0.2 ^b^	5.35 ± 0.01 ^c^	4.94 ± 0.05 ^d^
	2019	7.16 ± 0.03 ^a^	5.53 ± 0.2 ^b^	5.47 ± 0.01 ^b^	4.84 ± 0.03 ^c^
	2020	6.88 ± 0.02 ^a^	5.95 ± 0.2 ^b^	5.17 ± 0.01 ^c^	4.63 ± 0.03 ^d^
	2021	7.28 ± 0.02 ^a^	5.63 ± 0.2 ^b^	5.27 ± 0.02 ^c^	4.64 ± 0.02 ^d^
Albertów LDS_0.05 for distance_ = 0.207 LDS_0.05 for years_ = n.s.	2017	5.81 ± 0.02 ^a^	5.60 ± 0.3 ^a^	4.62 ± 0.02 ^b^	3.99 ± 0.01 ^c^
	2018	6.13 ± 0.03 ^a^	5.38 ± 0.3 ^b^	4.74 ± 0.02 ^c^	3.86 ± 0.02 ^d^
	2019	5.95 ± 0.04 ^a^	5.56 ± 0.2 ^b^	4.86 ± 0.02 ^c^	3.95 ± 0.01 ^d^
	2020	5.97 ± 0.01 ^a^	5.44 ± 0.4 ^b^	4.58 ± 0.01 ^c^	4.03 ± 0.04 ^d^
	2021	6.09 ± 0.04 ^a^	5.72 ± 0.2 ^b^	4.60 ± 0.01 ^c^	3.91 ± 0.02 ^d^

Explanation: ^a–d^—different letters indicate significant difference at *p* ≤ 0.05 (LDS for distance from the road edge); n.s.—not significant at *p* ≤ 0.05.

**Table 4 toxics-11-00329-t004:** Organic carbon content (TOC in g kg^−1^) in soils before the COVID-19 pandemic (2017–2019) [28] and during the pandemic (2020–2021).

Site	Years	Distance from the Road Edge [m]			
		5	20	50	100
Piekoszów LDS_0.05 for distance_ = 0.195 LDS_0.05 for years_ = n.s.	2017	6.41 ± 0.03 ^a^	6.90 ± 0.01 ^b^	6.96 ± 0.02 ^b^	7.26 ^c^
	2018	6.58 ± 0.01 ^a^	6.96 ± 0.01 ^b^	7.26 ± 0.01 ^c^	7.34 ^c^
	2019	6.62 ± 0.03 ^a^	6.70 ± 0.02 ^a^	7.34 ± 0.03 ^b^	7.43 ^b^
	2020	6.38 ± 0.02 ^a^	6.82 ± 0.02 ^b^	7.20 ± 0.01 ^c^	7.28 ^c^
	2021	6.51 ± 0.02 ^a^	6.78 ± 0.02 ^b^	7.12 ± 0.02 ^c^	7.30 ^c^
Marcinkowice LDS_0.05 for distance_ = 0.274 LDS_0.05 for years_ = n.s.	2017	7.10 ± 0.02 ^a^	7.68 ± 0.02 ^b^	9.47 ± 0.01 ^c^	9.78 ^d^
	2018	6.98 ± 0.02 ^a^	7.72 ± 0.01 ^b^	9.60 ± 0.00 ^c^	9.84 ^c^
	2019	7.12 ± 0.01 ^a^	7.34 ± 0.01 ^a^	9.82 ± 0.02 ^b^	9.86 ^b^
	2020	6.92 ± 0.02 ^a^	7.40 ± 0.01 ^b^	9.20 ± 0.02 ^c^	9.82 ^d^
	2021	7.08 ± 0.01 ^a^	7.65 ± 0.02 ^b^	9.44 ± 0.02 ^c^	9.90 ^d^
Skorzeszyce LDS_0.05 for distance_ = 0.186 LDS_0.05 for years_ = n.s.	2017	8.38 ± 0.02 ^a^	8.68 ± 0.02 ^b^	9.58 ± 0.02 ^c^	9.48 ^c^
	2018	8.45 ± 0.02 ^a^	8.80 ± 0.02 ^b^	9.52 ± 0.01 ^c^	9.58 ^c^
	2019	8.52 ± 0.02 ^a^	8.64 ± 0.00 ^a^	9.64 ± 0.01 ^b^	9.76 ^b^
	2020	8.29 ± 0.02 ^a^	8.52 ± 0.01 ^b^	9.41 ± 0.01 ^c^	9.52 ^c^
	2021	8.34 ± 0.02 ^a^	8.74 ± 0.01 ^b^	9.60 ± 0.03 ^c^	9.70 ^c^
Giełczew II LDS_0.05 for distance_ = 0.236 LDS_0.05 for years_ = n.s.	2017	8.10 ± 0.01 ^a^	8.28 ± 0.01 ^a^	9.20 ± 0.01 ^b^	9.24 ^b^
	2018	7.96 ± 0.02 ^a^	8.34 ± 0.02 ^b^	9.32 ± 0.00 ^c^	9.28 ^c^
	2019	7.98 ± 0.01 ^a^	8.04 ± 0.02 ^a^	8.92 ± 0.01 ^b^	8.94 ^b^
	2020	7.90 ± 0.01 ^a^	8.20 ± 0.00 ^b^	9.26 ± 0.02 ^c^	9.30 ^c^
	2021	7.95 ± 0.02 ^a^	8.26 ± 0.02 ^b^	9.04 ± 0.02 ^c^	9.12 ^c^
Łuszczów Kolonia LDS_0.05 for distance_ = 0.337 LDS_0.05 for years_ = n.s.	2017	7.76 ± 0.02 ^a^	8.37 ± 0.02 ^b^	9.40 ± 0.02 ^c^	9.86 ^d^
	2018	7.82 ± 0.02 ^a^	8.30 ± 0.01 ^b^	10.28 ± 0.02 ^c^	9.82 ^d^
	2019	7.91 ± 0.01 ^a^	8.36 ± 0.01 ^b^	9.96 ± 0.01 ^c^	9.97 ^c^
	2020	7.74 ± 0.02 ^a^	8.30 ± 0.01 ^b^	9.52 ± 0.02 ^c^	9.90 ^d^
	2021	7.80 ± 0.01 ^a^	8.24 ± 0.01 ^b^	9.96 ± 0.00 ^c^	9.88 ^c^
Albertów LDS_0.05 for distance_ = 0.206 LDS_0.05 for years_ = n.s.	2017	8.72 ± 0.02 ^a^	9.26 ± 0.02 ^b^	9.32 ± 0.03 ^b^	9.38 ^b^
	2018	8.74 ± 0.01 ^a^	8.91 ± 0.01 ^a^	9.50 ± 0.03 ^b^	9.32 ^b^
	2019	8.80 ± 0.01 ^a^	8.92 ± 0.02 ^a^	9.68 ± 0.01 ^b^	9.36 ^c^
	2020	8.68 ± 0.02 ^a^	8.84 ± 0.01 ^a^	9.45 ± 0.03 ^b^	9.40 ^b^
	2021	8.70 ± 0.01 ^a^	8.90 ± 0.00 ^a^	9.62 ± 0.01 ^b^	9.42 ^b^

Explanation: ^a–d^—different letters indicate significant difference at *p* ≤ 0.05 (LDS for distance from the road edge); n.s.—not significant at *p* ≤ 0.05.

**Table 5 toxics-11-00329-t005:** Total nitrogen content (TN in g kg^−1^) in soils before the COVID-19 pandemic (2017–2019) [28] and during the pandemic (2020–2021).

Site	Years	Distance from the Road Edge [m]			
		5	20	50	100
Piekoszów LDS_0.05 for distance_ = 0.033 LDS_0.05 for years_ = n.s.	2017	0.43 ± 0.02 ^a^	0.49 ± 0.02 ^b^	0.74 ± 0.01 ^c^	0.75 ± 0.01 ^c^
	2018	0.42 ± 0.01 ^a^	0.51 ± 0.01 ^b^	0.76 ± 0.02 ^c^	0.78 ± 0.02 ^c^
	2019	0.44 ± 0.02 ^a^	0.52 ± 0.02 ^b^	0.78 ± 0.01 ^c^	0.80 ± 0.01 ^c^
	2020	0.45 ± 0.01 ^a^	0.52 ± 0.02 ^b^	0.78 ± 0.02 ^c^	0.80 ± 0.02 ^c^
	2021	0.42 ± 0.01 ^a^	0.49 ± 0.01 ^b^	0.74 ± 0.01 ^c^	0.76 ± 0.01 ^c^
Marcinkowice LDS_0.05 for distance_ = 0.035 LDS_0.05 for years_ = n.s.	2017	0.62 ± 0.02 ^a^	0.66 ± 0.02 ^b^	1.02 ± 0.02 ^c^	1.04 ± 0.02 ^c^
	2018	0.64 ± 0.01 ^a^	0.68 ± 0.02 ^b^	1.04 ± 0.01 ^c^	1.06 ± 0.02 ^c^
	2019	0.65 ± 0.02 ^a^	0.69 ± 0.01 ^b^	1.06 ± 0.01 ^c^	1.08 ± 0.02 ^c^
	2020	0.66 ± 0.02 ^a^	0.67 ± 0.02 ^a^	1.05 ± 0.01 ^b^	1.08 ± 0.02 ^b^
	2021	0.62 ± 0.02 ^a^	0.64 ± 0.01 ^a^	1.02 ± 0.01 ^b^	1.04 ± 0.01 ^b^
Skorzeszyce LDS_0.05 for distance_ = 0.032 LDS_0.05 for years_ = n.s.	2017	0.69 ± 0.01 ^a^	0.76 ± 0.01 ^b^	1.02 ± 0.02 ^c^	1.04 ± 0.01 ^c^
	2018	0.69 ± 0.01 ^a^	0.80 ± 0.01 ^b^	1.04 ± 0.02 ^c^	1.06 ± 0.01 ^c^
	2019	0.67 ± 0.01 ^a^	0.82 ± 0.02 ^b^	1.05 ± 0.02 ^c^	1.08 ± 0.01 ^c^
	2020	0.70 ± 0.01 ^a^	0.81 ± 0.01 ^b^	1.04 ± 0.00 ^c^	1.08 ± 0.02 ^c^
	2021	0.67 ± 0.01 ^a^	0.78 ± 0.01 ^b^	1.02 ± 0.02 ^c^	1.04 ± 0.00 ^c^
Giełczew II LDS_0.05 for distance_ = 0.041 LDS_0.05 for years_ = n.s.	2017	0.70 ± 0.02 ^a^	0.72 ± 0.00 ^a^	0.96 ± 0.01 ^b^	0.98 ± 0.01 ^b^
	2018	0.68 ± 0.01 ^a^	0.75 ± 0.02 ^b^	0.90 ± 0.01 ^c^	0.95 ± 0.01 ^d^
	2019	0.64 ± 0.02 ^a^	0.70 ± 0.00 ^b^	0.94 ± 0.00 ^c^	0.92 ± 0.02 ^c^
	2020	0.69 ± 0.03 ^a^	0.72 ± 0.03 ^a^	0.94 ± 0.02 ^b^	0.98 ± 0.00 ^b^
	2021	0.65 ± 0.02 ^a^	0.70 ± 0.02 ^b^	0.92 ± 0.02 ^c^	0.97 ± 0.02 ^c^
Łuszczów Kolonia LDS_0.05 for distance_ = 0.046 LDS_0.05 for years_ = n.s.	2017	0.64 ± 0.02 ^a^	0.69 ± 0.02 ^b^	1.04 ± 0.01 ^c^	1.08 ± 0.03 ^c^
	2018	0.67 ± 0.01 ^a^	0.72 ± 0.01 ^b^	1.05 ± 0.01 ^c^	1.10 ± 0.02 ^d^
	2019	0.68 ± 0.02 ^a^	0.70 ± 0.00 ^a^	1.12 ± 0.02 ^b^	1.15 ± 0.02 ^b^
	2020	0.66 ± 0.01 ^a^	0.69 ± 0.02 ^a^	1.06 ± 0.01 ^b^	1.12 ± 0.03 ^c^
	2021	0.64 ± 0.02 ^a^	0.68 ± 0.01 ^a^	1.04 ± 0.01 ^b^	1.08 ± 0.02 ^b^
Albertów LDS_0.05 for distance_ = 0.045 LDS_0.05 for years_ = n.s.	2017	0.76 ± 0.01 ^a^	0.80 ± 0.01 ^a^	0.96 ± 0.01 ^b^	0.98 ± 0.02 ^b^
	2018	0.74 ± 0.01 ^a^	0.76 ± 0.01 ^a^	0.98 ± 0.01 ^b^	0.99 ± 0.02 ^b^
	2019	0.70 ± 0.03 ^a^	0.74 ± 0.01 ^a^	0.98 ± 0.01 ^b^	0.98 ± 0.02 ^b^
	2020	0.78 ± 0.01 ^a^	0.82 ± 0.02 ^a^	0.98 ± 0.00 ^b^	0.98 ± 0.02 ^b^
	2021	0.72 ± 0.01 ^a^	0.74 ± 0.01 ^a^	0.96 ± 0.02 ^b^	0.98 ± 0.01 ^b^

Explanation: ^a–d^—different letters indicate significant difference at *p* ≤ 0.05 (LDS for distance from the road edge); n.s.—not significant at *p* ≤ 0.05.

**Table 6 toxics-11-00329-t006:** Cadmium content (Cd in mgkg^−1^) in soils before the COVID-19 pandemic (2017–2019) [28,29] and during the pandemic (2020–2021).

Site	Years	Distance from the Road Edge [m]			
		5	20	50	100
Piekoszów LDS_0.05 for distance_ = 0.042 LDS_0.05 for years_ = n.s.	2017	0.45 ± 0.01 ^a^	0.38 ± 0.02 ^b^	0.31 ± 0.02 ^c^	0.30 ± 0.01 ^c^
	2018	0.46 ± 0.00 ^a^	0.39 ± 0.01 ^b^	0.34 ± 0.01 ^c^	0.29 ± 0.03 ^c^
	2019	0.50 ± 0.02 ^a^	0.41 ± 0.02 ^b^	0.37 ± 0.03 ^bc^	0.33 ± 0.01 ^c^
	2020	0.45 ± 0.02 ^a^	0.36 ± 0.03 ^b^	0.30 ± 0.01 ^c^	0.27 ± 0.01 ^c^
	2021	0.48 ± 0.01 ^a^	0.39 ± 0.01 ^b^	0.35 ± 0.00 ^bc^	0.32 ± 0.01 ^c^
Marcinkowice LDS_0.05 for distance_ = 0.035 LDS_0.05 for years_ = n.s.	2017	0.38 ± 0.03 ^a^	0.36 ± 0.01 ^a^	0.30 ± 0.01 ^b^	0.26 ± 0.03 ^c^
	2018	0.41 ± 0.01 ^a^	0.38 ± 0.02 ^a^	0.32 ± 0.01 ^b^	0.29 ± 0.02 ^b^
	2019	0.42 ± 0.01 ^a^	0.40 ± 0.00 ^a^	0.33 ± 0.02 ^b^	0.30 ± 0.01 ^b^
	2020	0.40 ± 0.01 ^a^	0.37 ± 0.01 ^a^	0.33 ± 0.02 ^b^	0.28 ± 0.00 ^c^
	2021	0.42 ± 0.02 ^a^	0.39 ± 0.01 ^ab^	0.36 ± 0.03 ^b^	0.32 ± 0.02 ^c^
Skorzeszyce LDS_0.05 for distance_ = 0.049 LDS_0.05 for years_ = 0.034	2017	0.34 ± 0.02 ^aAB^	0.31 ± 0.02 ^abA^	0.28 ± 0.02 ^bcA^	0.26 ± 0.00 ^cAB^
	2018	0.36 ± 0.04 ^aB^	0.33 ± 0.02 ^abAB^	0.30 ± 0.00 ^bcAB^	0.26 ± 0.02 ^cAB^
	2019	0.41 ± 0.01 ^aC^	0.35 ± 0.01 ^bB^	0.33 ± 0.01 ^bcB^	0.29 ± 0.02 ^cB^
	2020	0.31 ± 0.02 ^aA^	0.30 ± 0.02 ^aA^	0.28 ± 0.03 ^abA^	0.25 ± 0.01 ^cA^
	2021	0.39 ± 0.01 ^aBC^	0.36 ± 0.01 ^aB^	0.31 ± 0.00 ^bAB^	0.27 ± 0.02 ^cAB^
Giełczew II LDS_0.05 for distance_ = 0.037 LDS_0.05 for years_ = n.s.	2017	0.40 ± 0.02 ^a^	0.39 ± 0.03 ^a^	0.32 ± 0.02 ^b^	0.31 ± 0.01 ^b^
	2018	0.42 ± 0.01 ^a^	0.41 ± 0.01 ^a^	0.33 ± 0.03 ^b^	0.28 ± 0.01 ^c^
	2019	0.43 ± 0.02 ^a^	0.42 ± 0.00 ^a^	0.37 ± 0.00 ^b^	0.33 ± 0.02 ^c^
	2020	0.42 ± 0.01 ^a^	0.40 ± 0.01 ^a^	0.31 ± 0.00 ^b^	0.30 ± 0.01 ^b^
	2021	0.44 ± 0.01 ^a^	0.41 ± 0.00 ^ab^	0.38 ± 0.01 ^b^	0.32 ± 0.01 ^c^
Łuszczów Kolonia LDS_0.05 for distance_ = 0.044 LDS_0.05 for years_ = 0.028	2017	0.60 ± 0.03 ^aB^	0.58 ± 0.02 ^aA^	0.53 ± 0.02 ^bA^	0.52 ± 0.03 ^bA^
	2018	0.59 ± 0.02 ^aAB^	0.61 ± 0.02 ^aB^	0.54 ± 0.04 ^bA^	0.54 ± 0.02 ^bAB^
	2019	0.63 ± 0.02 ^aC^	0.62 ± 0.01 ^aB^	0.57 ± 0.02 ^bB^	0.56 ± 0.02 ^bB^
	2020	0.57 ± 0.01 ^aA^	0.55 ± 0.01 ^abC^	0.52 ± 0.02 ^bcA^	0.50 ± 0.01 ^cAC^
	2021	0.62 ± 0.02 ^aBC^	0.61 ± 0.01 ^aB^	0.56 ± 0.03 ^bC^	0.54 ± 0.02 ^cAB^
Albertów LDS_0.05 for distance_ = 0.043 LDS_0.05 for years_ = n.s.	2017	0.35 ± 0.01 ^a^	0.29 ± 0.01 ^b^	0.26 ± 0.02 ^bc^	0.24 ± 0.02 ^c^
	2018	0.37 ± 0.03 ^a^	0.33 ± 0.03 ^a^	0.28 ± 0.01 ^bc^	0.26 ± 0.02 ^c^
	2019	0.40 ± 0.01 ^a^	0.35 ± 0.03 ^b^	0.33 ± 0.02 ^b^	0.26 ± 0.00 ^c^
	2020	0.39 ± 0.00 ^a^	0.34 ± 0.02 ^ab^	0.30 ± 0.01 ^b^	0.25 ± 0.03 ^c^
	2021	0.42 ± 0.01 ^a^	0.36 ± 0.02 ^b^	0.29 ± 0.02 ^c^	0.27 ± 0.02 ^c^

Explanation: ^a–c^—different letters indicate significant difference at *p* ≤ 0.05 (LDS for distance from the road edge); ^A–C^—different letters indicate significant difference at *p* ≤ 0.05 (LDS for years 2017–2021); n.s.—not significant at *p* ≤ 0.05.

**Table 7 toxics-11-00329-t007:** Lead content (Pb in mgkg^−1^) in soils before the COVID-19 pandemic (2017–2019) [28,29] and during the pandemic (2020–2021).

Site	Years	Distance from the Road Edge [m]			
		5	20	50	100
Piekoszów LDS_0.05 for distance_ = 3.36 LDS_0.05 for years_ = n.s.	2017	24.3 ± 0.13 ^a^	23.9 ± 0.32 ^a^	19.4 ± 0.23 ^b^	18.6 ± 0.25 ^b^
	2018	23.9 ± 0.06 ^a^	20.5 ± 0.71 ^ab^	18.1 ± 0.10 ^ab^	17.6 ± 0.11 ^b^
	2019	24.1 ± 0.05 ^a^	19.1 ± 0.58 ^b^	17.6 ± 0.19 ^b^	16.5 ± 0.35 ^b^
	2020	22.8 ± 0.10 ^a^	18.6 ± 0.65 ^b^	17.5 ± 0.43 ^b^	16.3 ± 0.18 ^b^
	2021	21.5 ± 0.12 ^a^	18.9 ± 0.87 ^b^	17.2 ± 0.19 ^b^	16.0 ± 0.15 ^b^
Marcinkowice LDS_0.05 for distance_ = 3.58 LDS_0.05 for years_ = n.s.	2017	34.1 ± 0.04 ^a^	31.9 ± 0.57 ^a^	16.9 ± 0.31 ^b^	16.4 ± 0.30 ^b^
	2018	30.4 ± 0.07 ^a^	29.3 ± 0.58 ^a^	15.8 ± 0.26 ^b^	16.2 ± 0.20 ^b^
	2019	28.2 ± 0.07 ^a^	26.4 ± 0.50 ^a^	15.1 ± 0.23 ^b^	14.9 ± 0.19 ^b^
	2020	28.1 ± 0.10 ^a^	25.9 ± 0.19 ^a^	15.0 ± 0.41 ^b^	16.4 ± 0.22 ^b^
	2021	27.7 ± 0.21 ^a^	25.4 ± 0.32 ^a^	15.2 ± 0.18 ^b^	16.3 ± 0.21 ^b^
Skorzeszyce LDS_0.05 for distance_ = 3.58 LDS_0.05 for years_ = n.s.	2017	35.1 ± 0.22 ^a^	31.8 ± 0.48 ^a^	26.3 ± 0.23 ^b^	23.2 ± 0.18 ^b^
	2018	33.0 ± 0.15 ^a^	28.3 ± 0.56 ^b^	23.9 ± 0.48 ^c^	22.6 ± 0.10 ^c^
	2019	32.2 ± 0.24 ^a^	27.8 ± 0.59 ^b^	21.5 ± 0.40 ^c^	20.9 ± 0.45 ^c^
	2020	31.9 ± 0.45 ^a^	27.6 ± 0.53 ^b^	21.3 ± 0.34 ^c^	20.7 ± 0.66 ^c^
	2021	31.1 ± 0.26 ^a^	27.4 ± 0.37 ^a^	21.1 ± 0.25 ^b^	20.4 ± 0.23 ^b^
Giełczew II LDS_0.05 for distance_ = 3.61 LDS_0.05 for years_ = n.s.	2017	24.2 ± 0.34 ^a^	23.9 ± 0.23 ^a^	16.1 ± 0.05 ^b^	15.4 ± 0.51 ^b^
	2018	22.9 ± 0.19 ^a^	23.1 ± 0.29 ^a^	14.2 ± 0.26 ^b^	14.7 ± 0.23 ^b^
	2019	21.8 ± 0.08 ^a^	22.0 ± 0.41 ^a^	14.5 ± 0.37 ^b^	13.8 ± 0.52 ^b^
	2020	21.6 ± 0.15 ^a^	21.2 ± 0.26 ^a^	14.4 ± 0.37 ^b^	14.1 ± 0.22 ^b^
	2021	21.5 ± 0.36 ^a^	21.3 ± 0.31 ^a^	14.5 ± 0.29 ^b^	13.9 ± 0.31 ^b^
Łuszczów Kolonia LDS_0.05 for distance_ = 4.41 LDS_0.05 for years_ = n.s.	2017	32.9 ± 0.11 ^a^	23.2 ± 0.63 ^b^	20.6 ± 0.50 ^c^	20.3 ± 0.40 ^c^
	2018	30.5 ± 0.19 ^a^	21.6 ± 0.37 ^b^	17.2 ± 0.64 ^c^	15.4 ± 0.28 ^c^
	2019	29.2 ± 0.22 ^a^	20.9 ± 0.24 ^b^	16.1 ± 0.63 ^c^	14.8 ± 0.31 ^c^
	2020	28.3 ± 0.23 ^a^	20.2 ± 0.60 ^b^	15.9 ± 0.73 ^c^	14.6 ± 0.31 ^c^
	2021	28.1 ± 0.19 ^a^	20.3 ± 0.18 ^b^	15.8 ± 0.32 ^c^	14.5 ± 0.19 ^c^
Albertów LDS_0.05 for distance_ = 4.00 LDS_0.05 for years_ = n.s.	2017	33.2 ± 0.19 ^a^	30.4 ± 0.29 ^a^	23.5 ± 0.20 ^b^	23.9 ± 0.30 ^b^
	2018	30.4 ± 0.27 ^a^	27.8 ± 0.63 ^a^	21.1 ± 0.25 ^b^	23.2 ± 0.38 ^b^
	2019	28.5 ± 0.22 ^a^	25.3 ± 0.26 ^a^	20.9 ± 0.44 ^b^	19.6 ± 0.21 ^b^
	2020	27.9 ± 0.05 ^a^	25.9 ± 0.52 ^a^	20.8 ± 0.10 ^b^	19.4 ± 0.30 ^b^
	2021	28.3 ± 0.22 ^a^	25.6 ± 0.59 ^a^	20.5 ± 0.23 ^b^	19.3 ± 0.33 ^b^

Explanation: ^a–c^—different letters indicate significant difference at *p* ≤ 0.05 (LDS for distance from the road edge); n.s.—not significant at *p* ≤ 0.05.

**Table 8 toxics-11-00329-t008:** Σ14PAHs content (Σ14PAHs in (μgkg−1, μg∙kg−1) in soils before the pandemic (2017–2019) [28] and during the COVID-19 pandemic (2020–2021).

Site	Years	Distance from the Road Edge [m]			
		5	20	50	100
Piekoszów LDS_0.05 for distance_ = 101.10 LDS_0.05 for years_ = 113.68	2017	1107.1 ± 0.74 ^aAB^	1086.8 ± 1.29 ^aAB^	460.8 ± 1.09 ^bA^	438.6 ± 1.75 ^bA^
	2018	1163.1 ± 0.57 ^aB^	1128.9 ± 1.12 ^aB^	511.7 ± 0.82 ^bA^	482.9 ± 0.76 ^bA^
	2019	1195.9 ± 1.13 ^aB^	1155.1 ± 0.86 ^aB^	523.5 ± 0.81 ^bA^	495.1 ± 1.56 ^bA^
	2020	1029.9 ± 0.60 ^aA^	1001.6 ± 1.01 ^aA^	480.3 ± 0.94 ^bA^	422.4 ± 0.95 ^bA^
	2021	1203.1 ± 0.54 ^aB^	1192.0 ± 1.40 ^aB^	512.9 ± 1.37 ^bA^	469.3 ± 1.22 ^bA^
Marcinkowice LDS_0.05 for distance_ = 121.34 LDS_0.05 for years_ = 126.81	2017	1063.4 ± 0.97 ^aAB^	1044.6 ± 1.21 ^aA^	456.1 ± 0.58 ^bAB^	436.8 ± 0.79 ^bA^
	2018	1166.5 ± 0.73 ^aBC^	1116.7 ± 0.99 ^aA^	536.2 ± 0.71 ^bAB^	501.4 ± 1.12 ^bA^
	2019	1200.9 ± 0.59 ^aC^	1119.5 ± 1.01 ^aA^	535.5 ± 1.00 ^bAB^	489.8 ± 1.04 ^bA^
	2020	1005.2 ± 1.08 ^aA^	996.9 ± 1.06 ^aA^	411.4 ± 0.67 ^bA^	413.8 ± 0.93 ^bA^
	2021	1241.0 ± 0.57 ^aC^	1024.3 ± 0.77 ^bA^	552.1 ± 0.77 ^bB^	490.6 ± 0.85 ^bA^
Skorzeszyce LDS_0.05 for distance_ = 70.82 LDS_0.05 for years_ = n.s.	2017	1049.2 ± 0.63 ^aA^	1015.2 ± 0.97 ^aA^	440.9 ± 1.17 ^bA^	418.7 ± 1.04 ^bA^
	2018	1094.8 ± 0.50 ^aA^	1063.8 ± 0.65 ^aA^	489.0 ± 0.97 ^bA^	464.7 ± 0.75 ^bA^
	2019	1105.8 ± 0.47 ^aA^	1068.9 ± 0.75 ^aA^	478.9 ± 1.08 ^bA^	459.0 ± 1.18 ^bA^
	2020	988.5 ± 0.51 ^aA^	971.6 ± 0.76 ^aA^	433.7 ± 0.53 ^bA^	419.5 ± 0.94 ^bA^
	2021	1112.3 ± 1.39 ^aA^	1092.3 ± 1.09 ^a^	460.9 ± 1.12 ^bA^	424.7 ± 0.65 ^bA^
Giełczew II LDS_0.05 for distance_ = 130.11 LDS_0.05 for years_ = 142.73	2017	1055.5 ± 0.79 ^aAB^	1020.6 ± 1.46 ^aAB^	436.7 ± 0.89 ^bA^	421.5 ± 1.02 ^bA^
	2018	1155.2 ± 0.24 ^aAB^	1130.0 ± 0.57 ^aAB^	525.5 ± 0.95 ^bA^	498.0 ± 0.61 ^bA^
	2019	1202.2 ± 0.99 ^aB^	1165.8 ± 1.18 ^aB^	490.7 ± 0.53 ^bA^	463.1 ± 1.18 ^bA^
	2020	1018.7 ± 0.46 ^aA^	1004.5 ± 0.93 ^aA^	422.8 ± 1.13 ^bA^	418.5 ± 1.13 ^bA^
	2021	1268.4 ± 1.51 ^aB^	1190.2 ± 0.56 ^aB^	486.1 ± 0.84 ^bA^	453.2 ± 1.05 ^bA^
Łuszczów Kolonia LDS_0.05 for distance_ = 112.78 LDS_0.05 for years_ = 127.41	2017	1161.2 ± 1.27 ^aAB^	1130.7 ± 0.76 ^aAB^	487.4 ± 0.78 ^bA^	464.8 ± 1.54 ^bA^
	2018	1265.8 ± 0.92 ^aAB^	1230.4 ± 1.25 ^aAB^	581.8 ± 0.77 ^bA^	563.7 ± 0.93 ^bA^
	2019	1301.5 ± 1.56 ^aB^	1253.6 ± 0.77 ^aB^	582.3 ± 0.99 ^bA^	540.4 ± 1.34 ^bA^
	2020	1154.7 ± 0.95 ^aA^	1120.5 ± 0.61 ^aA^	491.6 ± 1.06 ^bA^	455.2 ± 0.87 ^bA^
	2021	1328.6 ± 0.82 ^aB^	1249.8 ± 0.54 ^aB^	573.2 ± 1.00 ^bA^	528.9 ± 1.27 ^bA^
Albertów LDS_0.05 for distance_ = 97.11 LDS_0.05 for years_ = n.s.	2017	1001.6 ± 0.72 ^aA^	957.8 ± 1.08 ^aA^	402.9 ± 0.65 ^bA^	378.9 ± 1.33 ^bA^
	2018	1066.3 ± 0.83 ^aA^	1024.1 ± 1.11 ^aA^	485.0 ± 0.84 ^bA^	448.4 ± 0.98 ^bA^
	2019	1097.9 ± 0.62 ^aA^	1040.2 ± 0.90 ^aA^	489.5 ± 0.94 ^bA^	466.9 ± 0.61 ^bA^
	2020	984.9 ± 0.77 ^aA^	922.1 ± 0.60 ^aA^	392.6 ± 0.82 ^bA^	349.4 ± 1.29 ^bA^
	2021	1106.2 ± 0.48 ^aA^	1054.8 ± 1.00 ^aA^	459.5 ± 1.00 ^bA^	472.3 ± 1.18 ^bA^

Explanation: ^a,b^—different letters indicate significant difference at *p* ≤ 0.05 (LDS for distance from the road edge); ^A–C^—different letters indicate significant difference at *p* ≤ 0.05 (LDS for years 2017–2021); n.s.—not significant at *p* ≤ 0.05.

**Table 9 toxics-11-00329-t009:** Dehydrogenase activity (ADh in mg TPF kg^−1^ h^−1^) in soils before the COVID-19 pandemic (2017–2019) [28] and during the pandemic (2020–2021).

Site	Years	Distance from the Road Edge [m]			
		5	20	50	100
Piekoszów LDS_0.05 for distance_ = 0.337 LDS_0.05 for years_ = 0.554	2017	0.99 ± 0.04 ^aA^	1.25 ± 0.01 ^aA^	2.41 ± 0.02 ^bB^	2.45 ± 0.03 ^bB^
	2018	0.93 ± 0.02 ^aA^	1.17 ± 0.02 ^aA^	2.20 ± 0.01 ^bAB^	2.28 ± 0.01 ^bAB^
	2019	0.88 ± 0.02 ^aA^	0.94 ± 0.03 ^aA^	1.84 ± 0.02 ^bA^	1.83 ± 0.02 ^bA^
	2020	1.58 ± 0.02 ^aB^	2.84 ± 0.02 ^bB^	3.11 ± 0.02 ^cC^	3.28 ± 0.02 ^cC^
	2021	0.92 ± 0.03 ^aA^	1.05 ± 0.03 ^aA^	1.97 ± 0.03 ^bAB^	2.03 ± 0.02 ^bAB^
Marcinkowice LDS_0.05 for distance_ = 0.370 LDS_0.05 for years_ = n.s.	2017	0.93 ± 0.02 ^a^	1.20 ± 0.06 ^b^	4.08 ± 0.01 ^c^	4.76 ± 0.03 ^d^
	2018	0.89 ± 0.01 ^a^	1.08 ± 0.01 ^a^	3.78 ± 0.02 ^b^	4.13 ± 0.02 ^c^
	2019	0.80 ± 0.02 ^a^	0.95 ± 0.02 ^a^	3.16 ± 0.02 ^b^	3.37 ± 0.02 ^b^
	1.41	1.62 ± 0.04 ^a^	3.46 ± 0.04 ^b^	4.79 ± 0.02	4.81 ± 0.01 ^c^
	0.90	0.98 ± 0.01 ^a^	2.52 ± 0.01 ^b^	4.54 ± 0.02 ^c^	4.70 ± 0.02 ^d^
Skorzeszyce LDS_0.05 for distance_ = 0.335 LDS_0.05 for years_ = 0.729	2017	1.90 ± 0.02 ^aAB^	1.96 ± 0.03 ^aB^	3.73 ± 0.02 ^bB^	3.78 ± 0.01 ^bB^
	2018	1.59 ± 0.01 ^aAB^	1.65 ± 0.02 ^aAB^	2.92 ± 0.01 ^bA^	3.15 ± 0.04 ^bAB^
	2019	1.35 ± 0.01 ^aA^	1.16 ± 0.02 ^aA^	2.60 ± 0.03 ^bA^	2.94 ± 0.02 ^cA^
	2020	2.17 ± 0.02 ^aB^	2.39 ± 0.08 ^bB^	3.65 ± 0.03 ^cB^	3.80 ± 0.02 ^cB^
	2021	1.20 ± 0.01 ^aA^	1.26 ± 0.03 ^aAB^	2.89 ± 0.02 ^bA^	2.97 ± 0.02 ^bA^
Giełczew II LDS_0.05 for distance_ = 0.192 LDS_0.05 for years_ = n.s.	2017	1.48 ± 0.02 ^a^	2.49 ± 0.02 ^b^	4.28 ± 0.02 ^c^	4.45 ± 0.03 ^c^
	2018	1.69 ± 0.02 ^a^	2.26 ± 0.02 ^b^	4.16 ± 0.01 ^c^	4.33 ± 0.02 ^c^
	2019	1.38 ± 0.01 ^a^	2.12 ± 0.02 ^b^	4.12 ± 0.03 ^c^	4.14 ± 0.01 ^c^
	2020	1.86 ± 0.01 ^a^	2.74 ± 0.02 ^b^	4.55 ± 0.03 ^c^	4.60 ± 0.02 ^c^
	2021	1.23 ± 0.02 ^a^	1.83 ± 0.01 ^b^	4.06 ± 0.02 ^c^	4.39 ± 0.02 ^d^
Łuszczów Kolonia LDS_0.05 for distance_ = 0.230 LDS_0.05 for years_ = n.s.	2017	1.05 ± 0.02 ^a^	1.76 ± 0.01 ^b^	4.05 ± 0.02 ^c^	4.76 ± 0.01 ^d^
	2018	0.99 ± 0.01 ^a^	1.52 ± 0.02 ^b^	4.10 ± 0.01 ^c^	4.52 ± 0.03 ^d^
	2019	0.94 ± 0.02 ^a^	1.43 ± 0.01 ^b^	3.74 ± 0.01 ^c^	4.05 ± 0.02 ^d^
	2020	1.64 ± 0.03 ^a^	2.18 ± 0.01 ^b^	4.30 ± 0.03 ^c^	4.85 ± 0.01 ^d^
	2021	1.03 ± 0.01 ^a^	1.26 ± 0.01 ^a^	4.01 ± 0.03 ^b^	4.29 ± 0.02 ^c^
Albertów LDS_0.05 for distance_ = 0.224 LDS_0.05 for years_ = n.s.	2017	1.92 ± 0.02 ^a^	2.01 ± 0.02 ^a^	3.87 ± 0.02 ^c^	4.56 ± 0.02 ^d^
	2018	1.82 ± 0.02 ^a^	1.88 ± 0.01 ^a^	3.65 ± 0.03 ^b^	4.38 ± 0.02 ^d^
	2019	1.47 ± 0.02 ^a^	1.70 ± 0.02 ^b^	3.22 ± 0.01 ^c^	4.30 ± 0.01 ^d^
	2020	2.09 ± 0.01 ^a^	2.54 ± 0.02 ^b^	3.92 ± 0.02 ^c^	4.67 ± 0.03 ^d^
	2021	1.62 ± 0.02 ^a^	1.78 ± 0.01 ^a^	3.06 ± 0.02 ^c^	4.41 ± 0.02 ^d^

Explanation: ^a–d^—different letters indicate significant difference at *p* ≤ 0.05 (LDS for distance from the road edge); ^A–C^—different letters indicate significant difference at *p* ≤ 0.05 (LDS for years 2017–2021); n.s.—not significant at *p* ≤ 0.05.

**Table 10 toxics-11-00329-t010:** Neutral phosphatase activity (APh in mmol PNP kg^−1^ h^−1^) in soils before the COVID-19 pandemic (2017–2019) [28] and during the pandemic (2020–2021).

Site	Years	Distance from the Road Edge [m]			
		5	20	50	100
Piekoszów LDS_0.05 for distance_ = 0.748 LDS_0.05 for years_ = 1.598	2017	7.80 ± 0.03 ^aA^	7.94 ± 0.03 ^aA^	10.33 ± 0.03 ^bAB^	12.38 ± 0.04 ^cAB^
	2018	7.52 ± 0.01 ^aA^	7.68 ± 0.03 ^aA^	9.54 ± 0.04 ^bA^	12.57 ± 0.07 ^cB^
	2019	6.84 ± 0.04 ^aA^	7.05 ± 0.04 ^aA^	9.03 ± 0.04 ^bA^	9.81 ± 0.08 ^cA^
	2020	8.02 ± 0.02 ^aA^	8.34 ± 0.02 ^aA^	11.77 ± 0.06 ^bB^	12.39 ± 0.05 ^cAB^
	2021	6.71 ± 0.02 ^aA^	6.93 ± 0.02 ^aA^	9.26 ± 0.06 ^bA^	10.11 ± 0.02 ^cAB^
Marcinkowice LDS_0.05 for distance_ = 0.612 LDS_0.05 for years_ = n.s.	2017	8.59 ± 0.02 ^a^	9.97 ± 0.04 ^b^	14.08 ± 0.06 ^c^	14.09 ± 0.05 ^c^
	2018	8.55 ± 0.02 ^a^	9.08 ± 0.04 ^a^	13.85 ± 0.07 ^b^	13.86 ± 0.04 ^b^
	2019	7.16 ± 0.02 ^a^	8.27 ± 0.01 ^b^	13.01 ± 0.02 ^c^	13.26 ± 0.03 ^c^
	2020	9.22 ± 0.03 ^a^	10.15 ± 0.03 ^b^	14.37 ± 0.03 ^c^	14.56 ± 0.04 ^c^
	2021	6.98 ± 0.03 ^a^	8.06 ± 0.02 ^b^	12.85 ± 0.03 ^c^	14.18 ± 0.05 ^d^
Skorzeszyce LDS_0.05 for distance_ = 0.828 LDS_0.05 for years_ = 1.505	2017	7.06 ± 0.02 ^aAB^	9.24 ± 0.03 ^bB^	11.45 ± 0.02 ^cAB^	11.99 ± 0.05 ^cB^
	2018	6.67 ± 0.01 ^aAB^	8.44 ± 0.04 ^bAB^	10.15 ± 0.03 ^cAB^	10.37 ± 0.04 ^cAB^
	2019	5.64 ± 0.02 ^aA^	8.18 ± 0.02 ^bAB^	9.24 ± 0.03 ^cA^	9.62 ± 0.03 ^cAB^
	2020	7.38 ± 0.02 ^aB^	9.71 ± 0.02 ^bB^	11.89 ± 0.03 ^cB^	12.04 ± 0.03 ^cB^
	2021	6.29 ± 0.01 ^aAB^	7.34 ± 0.03 ^bA^	9.05 ± 0.03 ^cA^	9.27 ± 0.04 ^cA^
Giełczew II LDS_0.05 for distance_ = 0.605 LDS_0.05 for years_ = 1.190	2017	10.24 ± 0.02 ^aB^	10.89 ± 0.03 ^bB^	12.79 ± 0.05 ^cA^	13.02 ± 0.04 ^cB^
	2018	10.11 ± 0.03 ^aAB^	10.58 ± 0.03 ^aB^	12.39 ± 0.05 ^bA^	12.67 ± 0.03 ^cAB^
	2019	9.20 ± 0.02 ^aAB^	9.69 ± 0.02 ^aAB^	12.15 ± 0.03 ^bA^	11.67 ± 0.04 ^bA^
	2020	11.35 ± 0.03 ^aC^	11.45 ± 0.02 ^aB^	12.88 ± 0.06 ^bA^	13.57 ± 0.05
	2021	8.96 ± 0.03 ^aA^	9.08 ± 0.03 ^aA^	11.95 ± 0.01 ^bA^	13.21 ± 0.03 ^cB^
Łuszczów Kolonia LDS_0.05 for distance_ = 0.724 LDS_0.05 for years_ = 1.833	2017	7.26 ± 0.02 ^aA^	9.23 ± 0.02 ^bB^	11.33 ± 0.03 ^bA^	12.26 ± 0.03 ^cA^
	2018	7.18 ± 0.03 ^aA^	8.20 ± 0.03 ^bAB^	11.70 ± 0.02 ^cA^	11.82 ± 0.06 ^cA^
	2019	6.55 ± 0.02 ^aA^	6.78 ± 0.06 ^aA^	10.56 ± 0.05 ^bA^	11.15 ± 0.06 ^bA^
	2020	8.10 ± 0.02 ^aA^	9.87 ± 0.04 ^bB^	11.91 ± 0.03 ^bA^	12.35 ± 0.05 ^bA^
	2021	6.35 ± 0.02 ^aA^	6.42 ± 0.01 ^aA^	10.28 ± 0.06 ^bA^	11.16 ± 0.05 ^cA^
Albertów LDS_0.05 for distance_ = 0.735 LDS_0.05 for years_ = 1.712	2017	7.26 ± 0.00 ^aAB^	8.65 ± 0.04 ^bBC^	10.14 ± 0.04 ^bA^	11.72 ± 0.05 ^cA^
	2018	6.35 ± 0.02 ^aAB^	8.34 ± 0.02 ^bB^	9.65 ± 0.03 ^cA^	11.23 ± 0.04 ^dA^
	2019	6.22 ± 0.03 ^aAB^	6.43 ± 0.03 ^aA^	9.81 ± 0.03 ^bA^	10.83 ± 0.05 ^cA^
	2020	7.80 ± 0.00 ^aB^	9.15 ± 0.02 ^bC^	11.96 ± 0.03 ^cB^	11.95 ± 0.04 ^cA^
	2021	6.02 ± 0.02 ^aA^	6.22 ± 0.02 ^aA^	9.54 ± 0.03 ^bA^	10.73 ± 0.04 ^cA^

Explanation: ^a–d^—different letters indicate significant difference at *p* ≤ 0.05 (LDS for distance from the road edge); ^A–C^—different letters indicate significant difference at *p* ≤ 0.05 (LDS for years 2017–2021); n.s.—not significant at *p* ≤ 0.05.

**Table 11 toxics-11-00329-t011:** Urease activity (AU in mg N-NH_4_^+^ kg^−1^ h^−1^) in soils before the COVID-19 pandemic (2017–2019) [28] and during the pandemic (2020–2021).

Site	Years	Distance from the Road Edge [m]			
		5	20	50	100
Piekoszów LDS_0.05 for distance_ = 0.318 LDS_0.05 for years_ = n.s.	2017	14.15 ± 0.03 ^a^	12.53 ± 0.01 ^b^	10.70 ± 0.06 ^c^	10.95 ± 0.06 ^c^
	2018	13.62 ± 0.02 ^a^	12.40 ± 0.03 ^b^	10.08 ± 0.01 ^c^	10.32 ± 0.04 ^c^
	2019	13.37 ± 0.02 ^a^	12.56 ± 0.02 ^b^	10.65 ± 0.01 ^c^	10.86 ± 0.01 ^c^
	2020	12.78 ± 0.01 ^a^	12.34 ± 0.02 ^b^	10.53 ± 0.02 ^c^	10.89 ± 0.02 ^d^
	2021	13.22 ± 0.02 ^a^	12.01 ± 0.01 ^b^	10.76 ± 0.03 ^c^	10.62 ± 0.02 ^c^
Marcinkowice LDS_0.05 for distance_ = 0.698 LDS_0.05 for years_ = 1.693	2017	15.53 ± 0.03 ^aB^	12.14 ± 0.02 ^bB^	10.39 ± 0.02 ^cB^	10.74 ± 0.01 ^cA^
	2018	14.61 ± 0.02 ^aAB^	10.59 ± 0.02 ^bA^	9.42 ± 0.01 ^cAB^	9.43 ± 0.02 ^cA^
	2019	13.95 ± 0.03 ^aAB^	9.62 ± 0.01 ^bA^	8.58 ± 0.01 ^cA^	8.40 ± 0.02 ^cA^
	2020	12.67 ± 0.02 ^aA^	11.86 ± 0.02 ^bB^	10.35 ± 0.02 ^cB^	10.61 ± 0.04 ^cA^
	2021	13.82 ± 0.02 ^aAB^	10.15 ± 0.02 ^bA^	9.20 ± 0.03 ^cAB^	9.86 ± 0.02 ^bcA^
Skorzeszyce LDS_0.05 for distance_ = 0.645 LDS_0.05 for years_ = 0.840	2017	12.42 ± 0.01 ^aB^	12.06 ± 0.02 ^abB^	11.54 ± 0.03 ^abB^	10.01 ± 0.03 ^cB^
	2018	11.68 ± 0.03 ^aAB^	11.05 ± 0.03 ^abA^	10.79 ± 0.04 ^bA^	9.63 ± 0.03 ^cAB^
	2019	11.25 ± 0.01 ^aA^	10.69 ± 0.01 ^abA^	10.44 ± 0.02 ^bA^	8.77 ± 0.02 ^cA^
	2020	12.46 ± 0.03 ^aB^	12.33 ± 0.03 ^abB^	12.09 ± 0.00 ^abB^	11.72 ± 0.02 ^bB^
	2021	12.27 ± 0.02 ^aB^	12.08 ± 0.01 ^aB^	11.38 ± 0.01 ^bB^	8.95 ± 0.03 ^cA^
Giełczew II LDS_0.05 for distance_ = 0.439 LDS_0.05 for years_ = n.s.	2017	16.38 ± 0.03 ^a^	14.83 ± 0.04 ^b^	10.36 ± 0.02 ^c^	10.68 ± 0.02 ^c^
	2018	15.65 ± 0.03 ^a^	14.39 ± 0.04 ^b^	9.48 ± 0.01 ^c^	10.22 ± 0.02 ^c^
	2019	15.00 ± 0.01 ^a^	14.36 ± 0.02 ^b^	9.45 ± 0.02 ^c^	10.20 ± 0.02 ^c^
	2020	14.18 ± 0.02 ^a^	13.20 ± 0.02 ^b^	9.36 ± 0.03 ^c^	9.90 ± 0.02 ^d^
	2021	14.97 ± 0.03 ^a^	13.88 ± 0.01 ^b^	9.52 ± 0.01 ^c^	10.16 ± 0.01 ^d^
Łuszczów Kolonia LDS_0.05 for distance_ = 0.467 LDS_0.05 for years_ = n.s.	2017	17.40 ± 0.01 ^a^	14.43 ± 0.04 ^b^	11.93 ± 0.01 ^c^	12.56 ± 0.03 ^d^
	2018	16.72 ± 0.02 ^a^	14.44 ± 0.03 ^b^	11.26 ± 0.02 ^c^	12.18 ± 0.01 ^d^
	2019	16.81 ± 0.02 ^a^	14.25 ± 0.00 ^b^	10.62 ± 0.01 ^c^	10.66 ± 0.01 ^c^
	2020	15.34 ± 0.03 ^a^	13.67 ± 0.02 ^b^	11.43 ± 0.02 ^c^	11.85 ± 0.02 ^c^
	2021	16.50 ± 0.02 ^a^	13.98 ± 0.02 ^b^	11.05 ± 0.02 ^c^	11.24 ± 0.02 ^c^
Albertów LDS_0.05 for distance_ = 0.498 LDS_0.05 for years_ = n.s.	2017	13.60 ± 0.02 ^a^	12.54 ± 0.01 ^b^	10.03 ± 0.02 ^c^	10.18 ± 0.02 ^c^
	2018	13.48 ± 0.02 ^a^	10.73 ± 0.04 ^b^	10.12 ± 0.01 ^c^	9.37 ± 0.02 ^d^
	2019	13.00 ± 0.02 ^a^	10.31 ± 0.02 ^b^	9.67 ± 0.03 ^c^	8.62 ± 0.02 ^d^
	2020	11.92 ± 0.01 ^a^	11.08 ± 0.02 ^b^	9.43 ± 0.02 ^c^	9.20 ± 0.04 ^c^
	2021	12.55 ± 0.02 ^a^	11.76 ± 0.04 ^b^	9.71 ± 0.02 ^c^	8.93 ± 0.01 ^d^

Explanation: ^a–d^—different letters indicate significant difference at *p* ≤ 0.05 (LDS for distance from the road edge); ^A,B^—different letters indicate significant difference at *p* ≤ 0.05 (LDS for years 2017–2021); n.s.—not significant at *p* ≤ 0.05.

**Table 12 toxics-11-00329-t012:** Significant correlation coefficients between studied soil parameters (N = 30).

Parameter	pH_KCl_	TOC	TN	Cd	Pb	14PAHs	ADh	APh	AU
pH_KCl_	-	−0.61 ***	−0.49 **	n.s.	n.s.	n.s.	−0.52 **	n.s.	n.s.
TOC		-	0.97 ***	n.s.	0.49 **	n.s.	0.60 ***	n.s.	n.s.
TN			-	n.s.	0.48 **	n.s.	0.60 **	n.s.	n.s.
Cd				-	−0.41 *	0.75 ***	n.s.	n.s.	0.72 ***
Pb					-	−0.47 **	n.s.	n.s.	n.s.
14PAHs						-	−0.40 *	n.s.	0.40 *
ADh							-	0.64 ***	n.s.
APh								-	n.s.
AU									-

Explanation: *** significant at α = 0.0001; ** significant at α = 0.001; * significant at α = 0.01; n.s.—not significant.

## Data Availability

Not applicable.

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
