# Peer review of "Impact of COVID-19 Pandemic Constraints on the Ecobiochemical Status of Cultivated Soils along Transportation Routes"

_toxics, 2023, doi:10.3390/toxics11040329_

Round 1

Reviewer 1 Report

In the paper entitled “Impact of COVID-19 Pandemic Constraints on the Ecobiochemical Status of Cultivated Soils Along Transportation Routes” the authors described the impact of reduced mobility of people due to pandemic conditions. The analyzes covered measurements of basic soil analyses, several enzymes, heavy metals and PAHs content.

Although the paper was written very clearly and interestingly, the results did not reveal new facts. It was expected that there will be a reduction in pollution due to reduced traffic intensity. And as the authors have shown, the reduction was instant, already in 2021, the measurements showed coincidences with the year before the Pandemic. So this reduction was just an incident, like many things during the Pandemic, but not a trend.

“Monitoring of cultivated soils showed that the parameters of cultivated soils varied primarily according to the distance from the edge of the roadway”. This is well known fact, not a conclusion of this research. There is a lot of such examples in the text.

The main complaint is the lack of essential facts, conclusions drawn on the basis of the results, regardless of their volume and thorough analysis, which is the good side  of this work. Everything concluded, more or less could be said without analysis.

Author Response

Response to REVIEWER 1 

Manuscript  ID ijerph-2227904

Reviewer 2 Report

Review – Zawierucha et al.

The investigators evaluated the consequences of reduced travel  (presumably Poland) on soil health indicators along transportation routes (roads). They observed a few effects that were limited to proximity to the roadways. Heavy metals Pb and Cd and residual PAHs were dominantly changed. These effects were likely due to roadway treatment, prior vehicular traffic, and PAHS in the case of reduced traffic. Overall the authors have done a reasonable job of explaining their results and presenting them. However, there are several examples in which the study design and its interpretation require more explanation.

For example:

1)     What was the basis/rationale for enzyme selection? The tested enzymes are all used in assessing soil quality, but there are other enzymes that could have been used. Urease in particular functions extracellularly so it may not be a good enzyme to use to observe direct effects on microbes.

2)     In looking at enzymes and microbial populations the crop and the sample time are critical. Neither of these are well described in the work.

Specific Comments

Table 7. Be consistent with decimal points

L. 280-81 What do you mean by ‘appendage?’

Paragraph l. 331-338 That being the case, then mineralization should strip inorganic N.

L. 385-389 What’s the point of this paragraph? The article does not delve in deep enough detail, nor does it have the sampling robustness, to start making claims about potential consequences for human health.

L. 404-405 Not entirely true, while urease is inducible, it is also a constitutive enzyme in various bacteria and fungi. This is not a good explanation for urease activity. Consider that urease is most active in an alkaline environment. So, in what buffer system (or none at all) were the urease assays conducted?

L. 407-408 I find the explanation for reduced enzyme activities speculative.

L. 453-459 Table 1 does not show a statistically significant effect.

L. 471-475 This is not a relevant conclusion for the object of the study.

L. 478-479 Correlation is not necessarily causation.

Author Response

Response to Reviewer 2 

Manuscript ID ijerph-2227904

Reviewer 3 Report

The concept of research is appealing and scientifically sound documenting the effect of traffic load on soil dynamics. Besides, few inclusions may further add the qualitative improvement in manuscript:

·      Abstract: written well and adequate

·      Introduction: adequate and written nicely establishing the connect with the hypothesis.

·      Materials and Methods: As the effect of traffic load is being assessed on the soil health on the crop lands alongside transport routes, hence the cropping history of the sampling site throughout the experimental duration must be listed. If cropping patters are different in various sampling sites, discuss their impact on changed soil dynamics as well (in discussion section).

·      Results and Discussion are adequate. As the soil carbon, nutrient dynamics and biochemical (Enzyme) activities are significantly influenced with the change in microbial dynamics. Discussion part may be added with interactive effect of pollutants due to traffic on the above soil dynamics.

Ø Conclusion: Adequate

Ø References: References are relevant to the manuscript.

Author Response

Response to Reviewier3

Manuscript ID ijerph 2227904

Round 2

Reviewer 1 Report

Dear Sirs,

After re-examination of the paper entitled “Impact of COVID-19 Pandemic Constraints on the Ecobiochemical Status of Cultivated Soils Along Transportation Routes”, I inform you that my opinion remains unchanged.

All the best

Author Response

Response to Reviewier 1

manuscript ID 2227904

Reviewer 2 Report

Still no discussion of the consequences for variable crops on enzyme activity, which is a crucial feature for year-to-year variability.

Author Response

Response to Rewiever 2

Manuscript ID 2227904
